# Amniotic Membrane Graft in the Management of Complex Vaginal Mesh Erosion

**DOI:** 10.3390/jcm9020356

**Published:** 2020-01-28

**Authors:** Hui-Hsuan Lau, Quan-Bin Jou, Wen-Chu Huang, Tsung-Hsien Su

**Affiliations:** 1Department of Medicine, Mackay Medical College, New Taipei 252, Taiwan; 2Division of Urogynecology, Department of Obstetrics and Gynecology, Mackay Memorial Hospital, Taipei 104, Taiwan; 3Mackay Medicine, Nursing and Management College, New Taipei 251, Taiwan; 4Department of Obstetrics and Gynecology, Hsinchu Mackay Memorial Hospital, Hsinchu 300, Taiwan

**Keywords:** allograft, amnion, complication, surgical mesh

## Abstract

Vaginal mesh erosion is a devastating complication after pelvic floor mesh surgery and it can be treated conservatively or with surgical revision. However, the management options following a failed primary revision or complex vaginal erosions are very limited. The aim of this study is to describe a novel treatment using an amniotic membrane as an inlay graft for such patients. Eight patients who failed conservative or primary surgical revision were enrolled. The complex erosions included vaginal agglutination, multiple vaginal erosions, recurrent erosions, and mesh cutting through the urethra. We used an amniotic membrane as a graft to cover the vaginal defect after partial excision of the mesh erosion and we describe the technique in this study. There were no intraoperative complications and none of the patients reported any further symptoms at a mean of 27 months follow-up. Only one patient had recurrent erosion, however, the erosion size was narrower and was subsequently successfully repaired. No further vaginal mesh erosions were noted in the other patients who all had good functional recovery. The use of an amniotic graft can be an economic and alternative method in the management of complex vaginal mesh erosions.

## 1. Introduction

Due to the rapidly aging population worldwide, pelvic organ prolapse and stress urinary incontinence are not uncommon disorders. With regards to prolapse repair, a Cochrane systematic review reported that polypropylene mesh had a superior anatomical cure rate and could reduce the patient’s awareness of prolapse compared with traditional repair methods for cystocele repair [1]. With regards to the management of stress urinary incontinence, a mid-urethral sling is the mainstay of surgical treatment [2]. Over the past decade, vaginal mesh devices have been widely used to repair pelvic floor disorders. However, significant concerns have been raised about the safety of vaginal prosthetic mesh [1,3,4]. The United States Food and Drug Administration (FDA) issued warnings in 2008 and 2011 about mesh-related complications, including mesh erosion, pain, infection, bleeding, dyspareunia, organ perforation, and urinary problems and even stated in 2011 that “serious complications associated with surgical mesh for transvaginal repair of pelvic organ prolapse are not rare” [3,4]. Of these adverse effects, mesh erosion is a devastating complication and, hence, the management of mesh erosion has become an important health issue. 

Conservative treatment with antibiotics and vaginal estrogen cream can be used as first-line treatment for smaller mesh erosions and it has been reported that conservative treatment is more suitable for patients with vaginal mesh erosions smaller than 0.5 cm [5]. However, surgical revision should be considered for those with bigger or multiple erosions. Most vaginal erosions can be successfully repaired by primary tissue revision, however, recurrent erosions still occur and need repeated revisions despite careful surgical repair [5]. Very few studies have reported on how to approach recurrent erosions, complex erosions such as multiple erosions, mesh with vaginal stenosis or agglutination, or large erosions. In addition, only a few case series studies have shared their experience of using the Martius graft (bulbocavernosus fat graft) for erosions which did not respond to conservative treatment [6,7]. These complex erosions are difficult to repair and are always frustrating both for the patients and health-care professionals. 

Human amniotic membrane has successfully been used as a natural wound dressing in tissue healing applications. Clinically, amniotic membrane has been used to successfully treat diabetic foot ulcers and various types of ophthalmic problems, including chemical or thermal burns, persistent corneal epithelial defects, and corneal ulcers [8,9]. The beneficial effect of amniotic membrane on wound, ulcer, and defect healing is due to enhanced epithelialization and anti-fibrotic, anti-inflammatory, anti-angiogenic, and anti-microbial effects [8,9]. Due to the beneficial effects of amniotic membrane on wound healing, the aim of this study is to evaluate the efficacy and feasibility of using it as an inlay graft for complex mesh erosions.

## 2. Materials and Methods

### 2.1. Patient Selection

Fifty-three patients who had or were referred for mesh erosions were enrolled in this study from January 2015 to July 2019. All of the patients were managed with either topical estrogen or primary surgical revision. Among them, 8 (15%) had multiple erosions, severe vaginal stenosis or scarring, mesh perforating to an adjacent organ, large erosions (> 1.5 cm), or recurrent erosions (failed primary revision). The patients were comprehensively counseled on the use of an amniotic graft to treat erosions or vaginal defects and follow-up was scheduled at 1 week, 1, 3, and 6 months and yearly thereafter. The possible risk of infectious disease transmission was explained and all patients gave informed consent. This study was conducted from 2015 to 2019 and was approved by the Institutional Review Board of the hospital (15MMHIS071e). 

### 2.2. Amniotic Membrane Preparation

Amniotic membrane was obtained from prospective donors undergoing Caesarean sections who were negative for communicable diseases including HIV, syphilis, and hepatitis B and C. The placenta was cleaned with balanced salt solution containing a cocktail of antibiotics (50 mg/mL penicillin, 50 µg/mL streptomycin, 100 mg/mL neomycin, and 2.5 mg/mL amphotericin B) under sterile conditions. The amnion was separated from the chorion by blunt dissection. The separated membranes were then cut into different sizes and placed on nitrocellulose paper strips with the epithelial side up. Dulbecco Modified Eagle’s Medium/glycerol (1:1) was used for cryopreservation and the tissues were frozen at −80°C until use. Each piece of the amniotic membrane was stored in a separate container in a tissue bank. All amniotic membrane samples were negative for fungi and aerobic and anaerobic organisms.

### 2.3. Surgical Techniques

All patients underwent excision of the exposed mesh erosion and removal of granulation or scar tissue via a vaginal route. Thawed amniotic grafts were placed epithelial side up and inlayed to cover the underlying defect. The graft was then sutured to the edge of the vaginal defect using interrupted 3-0 Vicryl^TM^ sutures and any excess graft was trimmed off (Figure 1). All procedures were performed by two experienced attending urogynecologists and all the removed tissue specimens were subjected to pathological examination. Before skin incision, one dose of parenteral cefmetazole was given as preoperative antibiotic prophylaxis. The Foley was removed after the operation. Each patient was discharged on the first postoperative morning and given a 7-day course of oral cefuroxime.

## 3. Results

The characteristics of the patients are shown in Table 1. The average size of the erosions was 15 mm (range: 5–25 mm) and after an average 27 months (range: 6–45 months) of follow-up, seven patients had been successfully treated. One patient had recurrent vaginal erosion, however, the size was narrowed down to 3 mm and easily repaired with no recurrence after 12 months of follow-up. The excision of mesh, amniotic inlay and wound healing are shown in Figure 2. All patients were satisfied with the surgical outcomes.

None of the patients had complications during the operation and the average surgical time was 32 minutes (range: 25–45 minutes). Four patients complained of temporary vaginal discharge after surgery. Before surgery, five patients were sexually active and two patients complained of dyspareunia; however, none of the patients complained of dyspareunia after the surgery. No immunosuppressants were given and no immune graft rejection was encountered. All had good functional recovery. Moreover, none of the patients had recurrent prolapse, stress incontinence, wound infection, or pelvic pain.

## 4. Discussion

The present study shows the feasibility of an amniotic graft for the management of complex vaginal mesh erosions. This procedure may be an economic and practical alternative with patients who fail conservative management, have difficulty in primary revision surgery, and those with recurrent erosions.

Several management strategies have been reported for mesh erosions, including conservative treatment, primary defect closure, and partial mesh or total prosthesis excision. Graft repair methods include vaginal mucosal flaps or bulbocavernosi fat grafts. However, there is currently no standardized approach due to the different clinical features of mesh erosions. Surgery is considered after failure of conservative management or for larger erosions. We previously reported that 36% (20/56) of women with erosions were successfully treated with conservative management while 64% (36/56) required subsequent surgical revision. Compared to those requiring surgery, conservative treatment was successful if the size of the erosion was smaller than 0.5 cm [5]. The NICE (National Institute for Health and Clinical Excellence) guidelines state that surgery is advised if the erosion is larger than 1 cm and that complex cases are best managed in tertiary referral centers [10]. However, the surgical options for complex erosions are very limited and mesh erosion still recurs even after the most careful repair. Apart from the surgical techniques used, this may be due to wound condition and comorbidities such as tissue over-tensioning, diabetes, hyperlipidemia, and hypertension, all of which can adversely affect wound healing. In the present study, 63% (5/8) of the patients had these comorbidities and all were successfully repaired with the use of amniotic grafts.

Amniotic membrane has anti-inflammatory, anti-scarring, and anti-immunogenic properties. In addition, it can provide a moist environment and is an excellent substrate for epithelial growth, both of which are helpful for wound healing. As a result, amniotic grafts are widely used in the management of chronic diabetic foot ulcers, various ophthalmologic problems, and wound dressings [8,9,11]. Amnion is dissected from chorion because chorion is a vascular outer membrane in contact with the uterine wall. Amniotic membrane is an avascular fetal membrane and acts as a nonimmunogenic barrier between the mother and fetus. Therefore, amniotic allograft tissue can be transplanted without rejection by the host. Amniotic membrane is composed of three layers: the epithelium, basement membrane, and stroma. Similar to the repair of corneal ulcers, we kept the epithelial side up when inlaying the amniotic graft on vaginal erosions. This is because the basement membrane of the amnion acts as a substrate for progenitor epithelial cell growth by enhancing cell clonogenicity and preventing apoptosis [11]. The other advantages of amniotic membrane are easy procurement and low production costs. As a result, amniotic membrane is an ideal tissue which can facilitate the growth and differentiation of epithelial cells [8,9,11]. The nature of amnion, therefore, makes it a feasible graft which can facilitate the healing of vaginal erosions. Another surgical technique, the Martius flap procedure, also has many advantages, including a short operative time, little morbidity, and improved wound healing. However, this technique requires a lateral incision on the labia majora which may cause bleeding, hematoma, or cosmetic problems. In addition, inadequate tunnel or pedicle dissection may cause over-tension or vascular deficiency and adversely affect healing. Amniotic membrane grafts are easy to directly apply at the edges of a wound without making another incision. However, there are also risks of using an amnion graft, including transmission of viral, bacterial, or fungal infections to the recipient if the donor is not adequately screened or if the graft is not processed or stored properly.

Various approaches for mesh revision surgery have been reported in the literature, including transvaginal, laparoscopic, endoscopic, and abdominal approaches. The choice of intervention is based on the surgeon’s preference and on erosion characteristics. In the current study, we chose vaginal route surgery because it is less invasive. There are several limitations. First, the sample size was small which is because most patients with mesh complications and erosions can successfully be treated with conservative treatment or primary revision. Second, the follow-up evaluation should have been standardized and done by a third party who was blinded to the procedure. Third, long-term follow-up is required. Although the mean follow-up period was over 2 years, long-term follow-up is needed to monitor vaginal erosions due to the possibility of recurrence. The strength of this study is that it is the first to report the successful results of using amniotic grafts in the management of vaginal mesh erosions. A randomized controlled study of patients who do and do not receive treatment with amniotic membranes is warranted to confirm the effect on wound healing in women with complex vaginal mesh erosions. This technique can provide a low-cost and rapid recovery method for patients with serious vaginal mesh erosions. Since mesh complications are a very important health issue, the results of the current study may provide valuable information for pelvic surgeons with regards to using amniotic membrane in mesh erosion repair.

## Figures and Tables

**Figure 1 jcm-09-00356-f001:**
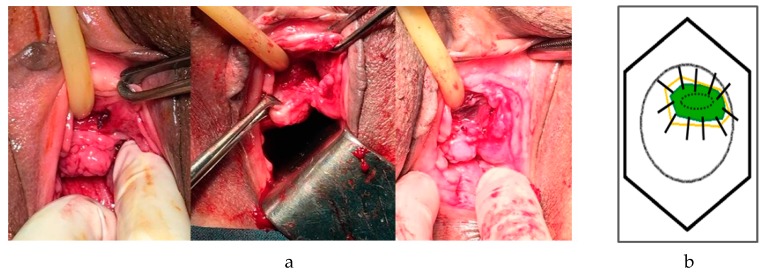
(**a**) Amniotic membrane used as an inlay graft. (**b**) Line diagram showing the amniotic membrane (solid orange lines) used as an inlay graft for a vaginal defect (green).

**Figure 2 jcm-09-00356-f002:**
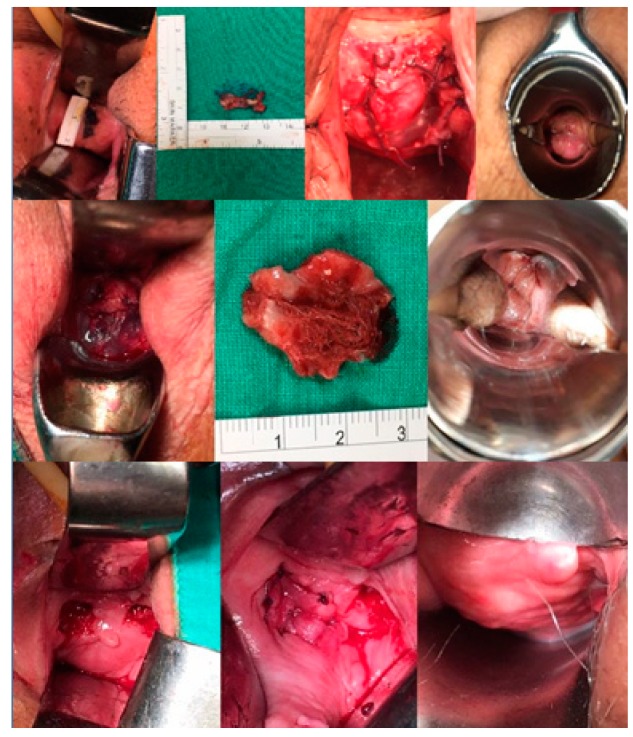
Amniotic graft inlay and wound healing of the patients.

**Table 1 jcm-09-00356-t001:** The characteristics of the patients.

No.	Age (year)	Parity	Mesh Kit	Recurrence (times)	Erosion		Comorbidity	Other problem	Surgical Treatment	Follow-up (months)
					Site	Size (cm)				
**1**	**62**	**3**	**Prolift**	**1**	**AVW APVW**	**2.5** **0.5**	**HTN**	No	EOME + AG	45
2	59	3	Perigee Apogee	No	Unknown	A vaginal dimple	No	Vaginal stenosis and severe scarring	Vaginoplasty and removal of part mesh + AG Vaginal dilator use	26
3	55	3	Prolift	No	AVW APVW	1.2 0.8	No	No	EOME + AG	34
4 *	57	2	2 times SL (unknown)	2	AVW	1.4	HPL	Tape cutting through urethra	EOME + AG.	34
5	83	3	Elevate	No	Vaginal cuff	1.8	HTN HPL	No	EOME + AG.	25
6	47	2	Unknown	1	AVW	1.5	No	No	EOME + AG.	21
7	76	2	Uphold	No	AVW	1.9	CAD, DM, HPL	No	EOME + AG	24
8	58	2	Uphold	No	AVW	1.5	DM, HTN	No	EOME + AG	6

Abbreviations. AVW: anterior vaginal wall; APVW: apical vaginal wall; SL: suburethral sling; CAD: coronary artery disease; DM: diabetes mellitus; HTN: hypertension; HPL: hyperlipidemia; EOME: excision of mesh erosion; AG: amniotic graft. * Recurrent erosion noted 12 months later with narrow down (3 mm) and repaired by surgical revision.

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
