# Peer review of "Amniotic Membrane Graft in the Management of Complex Vaginal Mesh Erosion"

_jcm, 2020, doi:10.3390/jcm9020356_

Round 1
Reviewer 1 Report
The study is quite interesting and well written. The procedure is clearly described, nevertheless I would suggest a major revision of the paper.
The Authors reported that this procedure is cheaper than others, can they give approximately the costs of the procedure? Did the patients need a post-surgical therapies? For how long did they use antibiotics? Did you have any graft rejection? What do you do to avoid graft rejection? The Authors chose a selected population affected by severe complications (stenosis, failure of repeated surgery, multiple erosions), can the Authors explain the reasons of this decision? Amnion graft is potential risk of transmission of viral infections, please comment more extensively this aspect The use of amniotic graft in creation of neovagina has been described in literature with contrasting results, only few Authors still use this technique because there are safer procedures. Please comment What are the advantages over others kind of graft? (for example “Martius" graft for the management of tension-free vaginal tape vaginal erosion”)Author Response
Please see the attachment.

Reviewer 2 Report
The authors describe a series of patients with difficult to cure mesh erosions treated by surgical resection and covering the wound by amniotic grafts that are sutured.
They claim complete healing in 7 cases and partial healing in 1. Patients were satisfied with the procedure and the healing process.
Follow up was 6 to 45 months.
The technique has not been applied for this indication yet and is therefore original. The text is concise and well written and the English language is perfect.
Authors should address following items.
Why is amnion sperated from chorion, wand exactly which layer is used? Why can entire choriamnion not be used? The beneficial effects (anti-inflammatory, tissue grow, anti-infectious…. are described. However, what is the evidence that this remains intact after freezing? This is a exploratory study of a smaal number of patients, without any control. As nobody knows how good a final resection and repair would do WITHOUT amniotic membrane in his particular set of patients, it cannot be stated that is a promising technique for the future. It should be stated that a randomized controlled study with and without amniontic membrane is warranted to really make sure the difference is due to the additional amniotic membrane. In such a study the follow up evaluation should be standardized and done by a third party who is blinded to the procedure. This should be made clear in the discussion and it it is essential in the conclusions, both in the text as well as in the abstract.Author Response
Please see the attachment.

Round 2
Reviewer 1 Report
The Authors have addressed my comments, however there are still some and syntax errors also in the new parts of the paper. The manuscript requires a more extensive revision of English.
Author Response
The manuscript was revised. A few mistakes have been corrected. A more extensive revision of English has been done.
